# Effects of Dietary Folic Acid Supplementation on Sex Differences in Oriental River Prawn, *Macrobrachium nipponense*

**DOI:** 10.3390/ani13233677

**Published:** 2023-11-28

**Authors:** Gang Jiang, Yucai Xue, Xuxiong Huang

**Affiliations:** 1Centre for Research on Environmental Ecology and Fish Nutrition (CREEFN) of the Ministry of Agriculture and Rural Affairs, Shanghai Ocean University, Shanghai 201306, China; jianggang940329@163.com (G.J.); d220100029@st.shou.edu.cn (Y.X.); 2Building of China—ASEAN Belt and Road Joint Laboratory on Mariculture Technology and Joint Research on Mariculture Technology, Shanghai 201306, China; 3National Demonstration Center for Experimental Fisheries Science Education, Shanghai Ocean University, Shanghai 201306, China

**Keywords:** *Macrobrachium nipponense*, folic acid, sex differences, growth performance, antioxidant activity

## Abstract

**Simple Summary:**

Understanding sex differences in the utilization of folic acid is essential for *Macrobrachium nipponense* in their feed formulation programs. In the present study, prawns were fed diets formulated with three folic acid levels to assess growth performance, fecundity, oxidation resistance, and the transcriptome difference between males and females. The current results indicated females appear to require more folic acid levels during the reproductive stage and have a better tolerance to thermal stress compared to males when fed with the same dietary folic acid levels.

**Abstract:**

To better understand the sex differences in the utilization of folic acid for male and female prawns, *Macrobrachium nipponense*, three experimental diets with graded folic acid contents (A: 1.22 mg/kg, B: 5.44 mg/kg, and C: 10.09 mg/kg) were provided to prawns for 8 weeks. The experimental study demonstrated that prawns fed diets containing 1.22 mg/kg showed the best values on body weight gain (BWG), specific growth rate (SGR), and individual weight for both males and females. Male prawns fed above 1.22 mg/kg of dietary folic acid significantly decreased the crude protein contents in their bodies (*p* < 0.05), while no significant differences were observed in females among all treatments (*p* > 0.05). The protease activity of the hepatopancreas in females was significantly (*p* < 0.05) improved by a dietary level of 10.09 mg/kg of folic acid. However, the opposite trend was observed in males, with the highest protease activity observed at a dietary level of 1.22 mg/kg folic acid. Increasing dietary folic acid levels did not suppress early maturation but led to an increase in the fecundity of females. Furthermore, prawns fed with 10.09 mg/kg of folic acid exhibited improved tolerance against thermal stress in both males and females. Transcriptome analysis revealed that during thermal stress, the “Oxidative phosphorylation” and “Pantothenate and CoA biosynthesis” signaling pathways were significantly enriched in females, and the “Fatty acid biosynthesis” signaling pathway was significantly enriched in males. The results of this study preliminarily evaluate the differences between male and female *M. nipponense* in response to different dietary folic acid levels and are helpful in promoting the health and growth of aquaculture production of this species.

## 1. Introduction

The oriental river prawn, *Macrobrachium nipponense*, belongs to the Palaemonidae family of decapod crustaceans. It is widely distributed in freshwater and low-salinity estuarine regions of East Asia [1,2]. Due to its remarkable adaptability and high nutritional value, *M. nipponense* has gained significant recognition as an important species for commercial aquaculture in China [3,4]. Similar to *Macrobrachium rosenbergii*, *M. nipponense* exhibits sexual dimorphism in growth patterns [5], with male prawns typically growing faster and reaching larger sizes at harvest time [6].

The absorption and nutrient requirements of aquatic animals are influenced by species [7], the physical–chemical environment [8], and the interactions with other dietary nutrients [9]. Wu et al. revealed that nutrient requirements and absorption capacities may differ between genders within the same species [10]. Julio et al. also confirmed this view, and the females of *Sardinops sagax*, *Euphylax dovii*, and *Stenocionops ovata* seem to require a greater amount and type of prey consumed than males [11]. Folic acid, with the chemical formula C_19_H_19_N_7_O_6_, is a water-soluble vitamin B_9_ [12]. It serves as an essential micro-nutrient for aquatic animals economically as a carbon group donor or acceptor [13]. Previous studies showed that adequate dietary folic acid intake can improve growth and reproduction performance, mediate intestinal microflora, enhance antioxidant capacity and immune response, and change alkaline phosphatase activity [14,15,16]. Furthermore, Annamalai et al. revealed that appropriate dietary folic acid levels can improve the muscle biochemical composition of *M. rosenbergii* [17]. Similar results can also be found in *Penaeus monodon* [18], *Fenneropenaeus chinensis* [19], and *Eriocheir sinensis* [20]. Therefore, it is particularly necessary to explore the adequate requirements of this vitamin for economically important species in aquaculture. Up to now, the quantitative requirements of folic acid have been reported in various prawns, including 2.0 mg/kg for *M. rosenbergii* [17], 1.9–2.1 mg/kg for *P. monodon* [18], and 5 mg/kg for *F. chinensis* [19]. Nevertheless, no research has been conducted on how dietary folic acid levels affect *M. nipponense*, especially with regard to gender specificity.

To better understand the effects of folic acid on male and female *M. nipponense*, prawns were reared on diets with different levels of supplemental folic acid. Then, the growth performance, prawn biochemical compositions, digestive enzyme activities, fecundity, antioxidant activity, and transcriptome analysis of male and female prawns were investigated. This research will provide new knowledge for understanding the difference in absorption and utilization of folic acid between genders in *M. nipponense*.

## 2. Materials and Methods

### 2.1. Test Prawn and Feeding Protocol

The broodstock of *M. nipponense* was based on the inbred offspring of a monophyletic sibling, as established by Jiang [21]. A berried female was incubated in a 5 L aquarium. Once hatched, all zoea larvae were transferred to and cultured in a larger polyethylene tank with a capacity of 10 L. Once the zoea larvae had completely metamorphosed into post-larvae stage, 480 healthy post-larvae zoea (with an initial average wet weight of 12 ± 1 mg) were randomly allocated into twelve experimental cages (0.5 m × 0.5 m × 1.2 m) involving three diet treatments with 40 prawns per cage in four replicates. The prawns were fed twice daily at 6:00 a.m. and 5:00 p.m. for 8 weeks, with a feeding rate of 5% of the biomass of the cage. Dechlorinated tap water was used to change one-third of the water in each cage every three days. Throughout the feeding trial period, *M. nipponense* was offered *Ceratophyllum demersum* L., an aquatic plant, as a protective habitat. The prawn culture maintained a water temperature of 31 ± 0.5 °C, pH levels between 7.5 and 8.0, and dissolved oxygen concentrations above 5 mg/L.

### 2.2. Diet Preparation

In the current study, the control diet (Table 1) was formulated based on folic acid requirements for shrimp [22]. Subsequently, three experimental diets were prepared by supplementing the control diet with commercial folic acid (Damasbeta, Shanghai Titan, China, purity: 99%), resulting in final folic acid contents of 1.22 (Treatment A), 5.44 (Treatment B), and 10.09 mg/kg (Treatment C), respectively, based on the analysis of the pellet diet [23]. To prepare the diet, the dry constituents were mechanically combined and incorporated into the lipid sources, resulting in a uniform blend. Distilled water was added in appropriate amounts to form a paste. After thoroughly blending, the paste was molded into pellets with a diameter of 1.0 mm and dried in open air for 24 h. The dried pellets were subsequently sealed in brown vacuum-packed plastic bags and stored at −20 °C until use.

### 2.3. Sample Collection and Biochemical Analysis

During the feeding trial period, the first spawning time of females in each treatment was recorded. Subsequently, a dissecting needle was used to peel off the eggs from the abdomen, and the number of eggs was counted under the microscope. The eggs were collected in a 1.5 mL EP pipe and stored at −20 °C until used for determining the proximate compositions. At the end of the feeding trial, all treatments fasted for 24 h before final sampling. The gender was distinguished by differences in secondary sexual characteristics, including physique, the second pereiopod, the width of the fifth pereiopod, the second ventral extremity, and the genital pore, between males and females. The details have been described by New et al. [23]. The total number and weight of males and females in each cage were subsequently recorded. Survivors and individual body weight were measured using the following formula:Survival rate (%) = (final number of prawn/initial number of prawn) × 100
Body weight gain (BWG, %) = [(final weight − initial weight)/initial weight] × 100
Specific growth rate (SGR, % day^−1^) = [(Ln final weight − Ln initial weight)/duration] ×100
Relative fecundity = the number of eggs/body weight of berried female (g)

Proximate compositions of experimental diets, prawn whole-body samples, and eggs were analyzed according to the standard methods of the Association of Official Analytical Chemists [24]. Moisture was determined via oven-drying at 105 °C to a constant weight. Ash was determined via incineration in a muffle furnace (SX-4-10, Shanghai Junzhuo, Shanghai, China) at 550 °C for 6 h. The crude protein contents of the diets and the whole body were determined using the Kjeldahl method (2300 Auto-analyzer, Foss Tecator). The total protein content of the eggs was determined using a kit (Jiancheng Biotech Co. Nanjing, China). Total lipid content in the diets, body samples, and eggs was measured using the method described by Cejas et al. (2004) [25] and Huang et al. [26].

The hepatopancreas from 5 male and female prawns in each tank were randomly selected and dissected. The samples were then weighed and homogenized with shrimp saline solution (28.4 g/L of NaCl, 1 g/L of MgCl∙6H_2_O, 2 g/L of MgSO_4_ ∙7H_2_O_2_, 2.25 g/L of CaCl_2_ ∙ 2H_2_O, 0.7 g/L of KCl, 1 g/L of glucose, and 2.38 g/L of Hepes) under ice-cooled conditions [27]. The supernatant fluid was collected after centrifugation (4000 rpm and 4 °C for 15 min) for the measurement of the digestive enzyme [28]. Kits obtained from Jiancheng Biotech Co. were utilized in the evaluation of amylase, lipase, and protease activities, following the manufacturer’s instructions.

### 2.4. Resistance of Prawn to the Thermal Stress

At the conclusion of the 8-week feeding trial, thirty male and female prawns were randomly selected from each treatment. These prawns were then utilized in triplicate to carry out a susceptibility assay to assess their response to thermal stress. The challenge test was performed in a 5 L aquarium and placed in the illumination incubators (GXZ-500B, Ningbo, China). The light intensity was set at 1000 lux with a light–dark cycle of 12:12 h. Based on pre-test results, the formal test stress temperature was set at 37 °C (the initial temperature of culture water was 31 °C, increasing by 1 °C every 24 h in the illumination incubators) for a duration of 48 h. After the stress assay, six living female and six living male prawns were randomly selected from the high concentration (Treatment C) and the low concentration (Treatment A) of dietary folic acid treatments. The hepatopancreas and gills were excised from these prawns for transcriptome sequencing. Next, the hepatopancreas of the remaining living female and male prawns from all three treatments were dissected to determine antioxidant enzyme activity. To ensure three biological replicates for each male and female sample, the tissue from two prawns was pooled together to form one biological replicate. All the tested tissues were immediately immersed in liquid nitrogen until further analysis.

For the antioxidant enzyme samples, the pretreatment method was identical to that described in Section 2.3 above. Superoxide dismutase (SOD), catalase (CAT), malondialdehyde (MDA), and total antioxidant capacity (T-AOC) were determined using kits (Jiancheng Biotech Co.) following the manufacturer’s instructions. For sample preparation in transcriptome sequencing, the hepatopancreas and gill tissues were homogenized using TRIzol reagent (Sangon Biotech, Shanghai, China) to extract total RNA, following the manufacturer’s instructions. The detailed experimental process can be found in our previously published study [21].

### 2.5. Transcriptomic Profiling Analysis

The Trinity software (Version 2.4.0) was used to assemble all the raw reads [29]. The NR protein, the GO, COG, and KEGG databases were then used to perform the gene annotation, using an E-value cut-off of 10^−5^ [30]. Differences in gene expressions of male and female prawns’ hepatopancreas and gills in the low and high folic acid-supplemented treatments were calculated using DESeq under the criteria of FDR (false discovery rate) < 0.05 [31].

### 2.6. Quantitative Real-Time PCR (qRT-PCR) Verification

The expression level of 10 DEGs was determined from the low and high folic acid supplementation treatments of male and female prawns’ hepatopancreas and gills by using quantitative real-time PCR (qRT-PCR) to verify the results of RNA-seq. Three biological replicates were measured for each sample. Designing the primers was facilitated with the Primer5 software (Version 5.0), while ß-actin served as the internal control (Appendix A). Experimental protocols and analysis methods were outlined in our previously published study [21].

### 2.7. Statistical Analysis

All data were presented as means ± standard deviation. The statistical analysis was performed using an analysis of variance (SPSS 25.0) after exploring the normality and homogeneity of the data. A natural log transformation was utilized to transfer non-normal distribution variables into normal distribution variables. Based on a one-way ANOVA, Duncan’s new multiple-range test was used to compare the differences among dietary treatments. Differences between treatments were considered significant when *p* < 0.05.

### 2.8. Ethics Approval

The experimental procedures conducted in this study strictly adhered to the ethical guidelines outlined in the EU Directive 2010/63/EU for animal experimentation. Approval was granted from the ethical guidelines of Shanghai Ocean University (SHOU-DW-2019-013) for the handling and utilization of experimental animals in the course of the experimental protocols.

## 3. Results

### 3.1. Growth Performance and Sex Differentiation

The body weight gain (BWG) and individual weight displayed the highest values in treatment A for both males and females, which were significantly different (*p* < 0.05) from those in treatments B and C (Figure 1). However, no significant difference was detected between treatments B and C (*p* > 0.05). The specific growth rate (SGR) in treatment A was significantly higher (*p* < 0.05) than that in treatment C. Nevertheless, there was no significant difference (*p* > 0.05) observed between treatments A and B, as well as treatments B and C, for both males and females. Additionally, there were no significant differences in survival rate (SR) or sex ratio among all treatments (*p* > 0.05) (Table 2).

### 3.2. Biochemical Analysis of Prawn

In males, there were no significant differences (*p* > 0.05) observed among all treatments for moisture and ash (Table 3). The crude protein content in treatment A was significantly higher (*p* > 0.05) than that in treatments B and C, whereas no significant difference (*p* > 0.05) was detected between treatments B and C. Additionally, the total lipid contents in treatments A and B did not exhibit a significant difference (*p* > 0.05). However, both were significantly higher (*p* < 0.05) than that in treatment C. In females, the total lipid content in treatment A was significantly higher than that in treatments B and C (*p* < 0.05). No significant differences (*p* > 0.05) were observed among all treatments for moisture, ash, and crude protein.

### 3.3. Digestive Enzyme Activities of Hepatopancreas

The activity of amylase, lipase, and protease was significantly influenced by high dietary folic acid levels in both males and females (Table 4). In males, the amylase activity in treatment A exhibited the highest value, which differed significantly (*p* < 0.05) from treatments B and C. Conversely, no significant differences were observed among all treatments in females (*p* > 0.05). The lipase activity in both males and females was significantly higher (*p* < 0.05) in treatment A than that in treatments B and C. However, no significant differences (*p* > 0.05) were found between treatments B and C. Regarding protease activity values, they increased first and then decreased in males, with treatment B exhibiting significantly higher values compared to treatments A and C (*p* < 0.05). Conversely, in females, the protease activity value continuously increased, with treatment C showing a significantly higher value than treatments A and B (*p* < 0.05).

### 3.4. The First Spawning Time and Fecundity

No significant difference (*p* > 0.05) was detected for the first spawning time of females among all treatments throughout the feeding trial (Figure 2). The number of eggs in treatment C was significantly higher (*p* < 0.05) than that in treatment A (Figure 3). However, no significant differences (*p* > 0.05) were detected between treatments A and B, as well as treatments B and C. Additionally, the relative fecundity in treatment A was significantly lower (*p* < 0.05) compared to treatments B and C (Figure 4). Nevertheless, no significant difference (*p* > 0.05) was detected between treatments B and C.

The dietary folic acid levels had significant effects (*p* < 0.05) on the total protein and total lipid contents of the eggs (Table 5). As the dietary folic acid levels increased, the total protein and total lipid contents of the eggs significantly increased (*p* < 0.05), and treatment C exhibited the highest values compared to treatments A and B. No significant differences were observed among all treatments for the moisture content of eggs (*p* > 0.05).

### 3.5. Survival and Antioxidant Activity in Hepatopancreas upon a Challenge with Thermal Stress

Generally, the cumulative survival of males and females in treatment C was significantly higher (*p* < 0.05) than that in treatment A (Figure 5). However, no significant differences (*p* > 0.05) were observed between treatments A and B, as well as treatments B and C, for both males and females.

The antioxidant enzyme activities in the hepatopancreas of male and female *M. nipponense* after exposure to thermal stress were significantly influenced by dietary folic acid levels (Table 6). In males, the activities of T-AOC, SOD, and CAT in treatment C were all significantly higher (*p* < 0.05) than those in treatment A. Furthermore, the T-AOC activity in treatment C was significantly higher (*p* < 0.05) than that in treatment B. However, no significant difference (*p* > 0.05) was observed for SOD activity between treatments B and C. Conversely, the MDA activity significantly increased (*p* < 0.05) with higher dietary folic acid levels. In females, the T-AOC activity significantly increased (*p* < 0.05) with higher dietary folic acid levels. The SOD activity in treatment A was significantly lower (*p* < 0.05) than that in treatments B and C. However, there was no significant (*p* > 0.05) difference between treatments B and C. No significant difference (*p* > 0.05) was detected in CAT activity among all treatments. Moreover, there was no significant difference (*p* > 0.05) in MDA activity between treatments A and B. However, in treatment C, it was significantly lower (*p* < 0.05) compared to the other two treatments.

### 3.6. Transcriptome Analysis

A total of 61,649 transcripts were obtained from male and female prawns fed with low (1.22 mg/kg) and high (10.09 mg/kg) folic acid levels. After removing low-quality reads, de novo assembly of transcripts resulted in 43,115 unigenes with an average length of 1024.36 bp. The length distribution of the unigenes was as follows: 401–600 bp (20.32%), 201–400 bp (19.38%), and >1800 bp (14.64%).

All 43,115 unigenes were annotated based on the *M. nipponense* genome [32]. A total of 36,596 (84.88%) unigenes matched known sequences in the *M.nipponense* genome. To identify putative functions, all the unigenes were blasted against six protein databases for annotation analysis, including NR, SwissProt, COG, KEGG, GO, and PFAM (Table 7). The NR database provided the highest number of gene annotations (39,753, 92.2%), followed by the COG (33,514, 77.73%) and the PFAM databases (28,185, 65.37%). The KEGG databases yielded the lowest unigene annotations (13,145, 30.49%) compared to the other five databases.

GO, COG, and KEGG analyses were performed to categorize and describe the gene products. In GO analysis, the matched unigenes fell into 51 functional groups, including three categories: biological process (23 functional groups, 10,640 unigenes), cellular component (15 functional groups, 10,700 unigenes), and molecular function (15 functional groups, 20,184 unigenes). The number of unigenes in each functional category significantly exceeded that identified with a GO analysis. This suggested that individual unigenes were assigned to multiple functional groups. The largest functional groups were “Binding” (14,892 unigenes), “Catalytic activity” (11,095 unigenes), “Cellular process” (8794 unigenes), “Metabolic process” (7847 unigenes), and “Cell part” (6584 unigenes) (Figure 6). In the COG database, the matched unigenes were classified into 22 functional categories. The largest groups, with more than 1000 unigenes each, were “Function unknown” (21,224 unigenes), “Replication, recombination and repair” (11,529 unigenes), “Posttranslational modification, protein turnover, chaperones” (1386 unigenes), and “Intracellular trafficking, secretion, and vesicular transport” (1377 unigenes) (Figure 7). KEGG analysis was applied to match the assembled unigenes to biological pathways. A total of 13,145 (30.49%) unigenes showed high similarity to known genes in the KEGG database. These unigenes were involved in various biological pathways, including “cellular processes” 2374 (18.06%), “environmental information processing” 3948 (30.03%), “genetic information processing” 1542 (11.73%), “human disease” 10,750 (81.78%), “metabolism” 3584 (27.27%), and “organismal systems” 6928 (52.70%) (Figure 8).

### 3.7. Analysis of the Intrinsic Regulatory Process of Folic Acid in Male and Female Prawns under Thermal Stress

Transcriptomic profiling analysis was conducted on the gills and hepatopancreas of male and female prawns of *M. nipponense* to reveal the intrinsic regulatory process. Compared to treatment A, the differentially expressed genes (DEGs) in both male and female prawns under treatment C were mainly downregulated (Figure 9). Between male gills in treatment A (LMG) and treatment C (HMG), a total of 543 unigenes exhibited differential expression, including 90 upregulated genes and 453 downregulated genes. Similarly, between female gills in treatment A (LFG) and treatment C (HFG), 906 unigenes displayed differential expression, with 700 upregulated genes and 206 downregulated genes. In male hepatopancreas, 377 unigenes displayed differential expression between treatment A (LMH) and treatment C (HMH), comprising 141 upregulated genes and 236 downregulated genes. Furthermore, between female hepatopancreas in treatment A (LFH) and treatment C (HFH), 359 unigenes were differentially expressed, with 142 upregulated genes and 217 downregulated genes.

KEGG analyses revealed that the “Lysosome”, “Glycolysis/Gluconeogenesis”, “PI3K-Akt signaling pathway”, and “Pyruvate metabolism” were significantly enriched pathways based on HFG_vs._LFG, HFH_vs._LFH, HMG_vs._LMG, and HMH_vs._LMH, respectively. Additionally, “Oxidative phosphorylation” and “Pantothenate and CoA biosynthesis” were enriched in HFG_vs._LFG and HFH_vs._LFH. The “Fatty acid biosynthesis” pathway was enriched in HMG_vs._LMG and HMH_vs._LMH (See Figure 10 below). These results suggested that the DEGs in these signaling pathways may influence the intrinsic regulatory process of folic acid in male and female *M. nipponense* under thermal stress. Twenty-five DEGs were screened out from these pathways, which were significantly differentially expressed in the male and female prawns under a high supplemental folic acid diet treatment (Table 8).

### 3.8. qRT-PCR Validation

To validate the reliability of the expression of DEGs identified via RNA-seq, 10 genes were randomly selected for validation using RT-qPCR (Figure 11). The qPCR results exhibited consistent expression patterns with those obtained from RNA-seq analysis. This confirmed the accuracy and reliability of the RNA-seq experiments.

## 4. Discussion

Previous research has demonstrated that folic acid supplementation can improve growth and health status in most aquatic animals, including improvements in growth rates, conversion efficiency, hematological parameters, and hepatosomatic indexes [17,33]. Nonetheless, the prawns fed with high folic acid levels showed poor performance in the current study. This study revealed that prawns fed with 1.22 mg/kg of folic acid level performed better growth performance than the other treatments, both in males and females (Table 2 and Figure 1). Similar results were also observed in *M. rosenbergii* [17] and the Chinese mitten crab *Eriocheir sinensis* [20]. The causes of this phenomenon may be attributed to the excessive supplementation of dietary folic acid, which could inhibit the expression of folic-acid-binding protein in *M. nipponense*. As a result, folic acid transport and metabolism are disrupted, leading to the inhibition of purine and pyrimidine synthesis, ultimately affecting protein synthesis. Notably, this speculation aligns with the findings of our study (Table 3). More interestingly, this study found that males exhibited better growth performances compared to females among all dietary folic acid levels, potentially due to the sexually dimorphic growth pattern of *M. nipponense* [5].

Crustaceans heavily rely on their digestive enzymes to effectively control and optimize the process of nutrient digestion and absorption, which ultimately impacts their growth rate [17,33,34]. By enhancing the activity of these enzymes, the organism’s ability to digest food and utilize nutrients can be significantly improved [35]. Interestingly, the levels of digestive enzyme activity in the hepatopancreas exhibited a non-linear relationship with the dietary supplementation of folic acid in both male and female *M. nipponense* individuals. In this study, when the dietary folic acid content exceeded 1.22 mg/kg, lipase activities in both female and male prawns decreased. These results are consistent with the research on *M. rosenbergii*, which also demonstrated a decrease in digestive enzyme activities due to high dietary folic acid levels [17]. Furthermore, excessive intake of folic acid, according to Xue et al., can lead to an increase in DNA methylation levels and lipoprotein lipase, resulting in a decrease in triglyceride lipase activities in the hepatopancreas [36]. This, in turn, affects digestion and feeding capacity. Furthermore, in male hepatopancreas, the activity of amylase decreased as dietary folic acid levels increased. However, no significant difference was detected among all treatments in females. In general, amylase breaks down starch or carbohydrates into glucose [37]. Therefore, the possible cause of this phenomenon could be attributed to the potential variation in the ability of starches or carbohydrates to break down into glucose among different sexes of crustaceans. Moreover, protease activity was highest in treatment B for males but increased with dietary folic acid levels for females. Yamashita and Konagaya et al. discovered an increase in protease activity exclusively in female *Ayu Plecoglossus altivelis* during the spawning season, while there were no changes in the male fish [38]. Therefore, we speculated that dietary folic acid may impact the fecundity of *M. nipponense*. Furthermore, the divergent folic acid requirements between males and females during the breeding season could potentially contribute to the observed variations in protease activity within the hepatopancreas. Consequently, compared with male prawns, female prawns appeared to require higher dietary folic acid levels to maintain the hepatopancreas’ ability to absorb and utilize food.

Although high dietary folic acid supplementation does not change the sex ratio of *M. nipponense* in the population, the role of folic acid in improving reproductive performance should not be ignored. Numerous studies have demonstrated that folic acid improves fertility by reducing zygote mortality and increasing both egg number and offspring survival [39,40]. The results of the current study also verified this viewpoint (Figure 2, Figure 3 and Figure 4). As the quantities of folic acid in the diet increased, the spawning time of females in each treatment advanced, even though there were no considerable variations detected among all treatments. However, the amount of egg holding and relative fecundity were significantly higher in the high folic acid supplemental treatments (treatment C) than in treatment A. This suggests that extra dietary folic acid during reproduction has the potential to improve the overall reproductive capabilities of females. The nutritional composition of eggs and offspring in decapod crustaceans appears to be intricately associated with their egg nourishment. [36]. Thus, the biochemical composition of eggs reflects the physiological state of the broodstock [41]. Triacylglycerol works as the most important biochemical compound for energy storage [42], and lipid levels in the eggs can reflect the nutritional indicators of spawners [43]. In our study, as dietary folic acid levels increased, the total lipid content of female eggs also significantly increased. This suggests that a greater spawning quantity corresponds to a higher energy requirement for spawners, ultimately resulting in improved reproductive performance.

Folic acid has been shown to enhance the antioxidant capacity of aquatic animals [17,33,44]. In this study, the addition of folic acid substantially enhanced the antioxidative defense mechanisms in both male and female prawns, including superoxide dismutase (SOD), catalase (CAT), and total antioxidant capacity (T-AOC). (Figure 6). Similar results were also observed in *M Rosenbergii* [17], *Eriocheir sinensis* [20], *Apostichopus japonicus* [45], and *Micropterus salmoides* [46]. The reactive species scavengers SOD and CAT play a crucial role in protecting cells from damage [47]. Maintaining a dynamic balance between normal metabolism and the production and elimination of reactive oxygen species (ROS) is essential. Antioxidant enzymes, such as SOD and CAT, effectively eliminate excessive ROS, thereby mitigating the harm caused by lipid peroxidation [48]. Consequently, these overdoses can be responsible for the increase in antioxidant enzyme activity in *M. nipponense* after thermal stress. MDA, the final product of lipid peroxidation, accumulates in cells and causes cytotoxicity, leading to damage to cells and tissues [49,50]. Significantly decreased hepatopancreatic MDA values were observed with increasing folic acid levels, with the highest values noted in male and female prawns treated with 1.22 mg/kg of folic acid (Treatment A) (Figure 6). These results suggest that high dietary folic acid intake may enhance lipid deposition and lipid peroxidation in male and female prawns. Moreover, prawns exposed to high dietary folic acid levels (treatment C) exhibited improved survival rates for males and females after thermal stress compared to those in the control treatment (treatment A) (Figure 5). These findings indicate that prawns fed with the basal diet experienced severe oxidative stress, accompanied by suppressed antioxidant responses. However, there is a scarcity of information concerning the antioxidant role played by folic acid in *M. nipponense*. Consequently, the mechanisms underlying the enhancement of prawns’ antioxidant capacity through the consumption of dietary folic acid need to be further clarified and elucidated.

Regulation of energy metabolism is usually adaptive to external environmental stress in aquatic animals [51,52]. Crustaceans consume additional energy to maintain a stable internal environment when faced with adverse environmental stress [53]. Transcriptome analyses revealed sex-specific differences in enriched pathways following thermal stress. In females, “Oxidative phosphorylation” and “Pantothenate and CoA biosynthesis” were all enriched based on HFG_vs._LFG and HFH_vs._LFH. “Oxidative phosphorylation” is critical for providing energy to aquatic animals in a stressful environment [54], and it is mediated via multi-subunit complexes. In this study, F-type H+-transporting ATPase subunit 6 (ATP5J) and aldehyde dehydrogenase (NAD+) (ALDH) were significantly upregulated in females. Both ATP5J and ALDH play a vital role in animal respiration and metabolism. This suggests that female prawns are capable of metabolizing more energy against heat stress when fed a high-concentration folic acid diet. This may also be the reason why females exhibited higher antioxidant levels and survival rates compared to males when subjected to the same dietary folic acid level and thermal stress conditions (Figure 5). Furthermore, female prawns fed with a high-concentration folic acid diet showed activation of the pantothenate and CoA biosynthesis pathways under individual thermal stress conditions. This pathway is crucial for the synthesis of CoA, which holds great significance in energy-related metabolism and various biological processes [55]. Thus, these findings further support the idea that female prawns fed with high folic acid demonstrate better resistance to thermal stress. On the other hand, in males, only the “Fatty acid biosynthesis” pathway was enriched based on HMG_vs._LMG and HMH_vs._LMH. Genes associated with this pathway, including fatty acid synthase, animal type (FASN), acetyl-CoA carboxylase/biotin carboxylase 1 (ACACA), and long-chain-fatty-acid-CoA ligase ACSBG (ACSBG), were all down-regulated. When organisms experience stress, they often down-regulate genes associated with metabolism and redirect their energy usage from reproduction to survival and homeostasis maintenance [56,57,58]. Reduced metabolism seems to be a response of male prawns to thermal stress. Therefore, these results provide evidence of the internal regulatory mechanism in male and female prawns of *M. nipponense* under thermal stress.

## 5. Conclusions

The present findings suggest that 1.22 mg/kg of dietary folic acid could meet the basic growth requirements for both male and female *M. nipponense*. However, it is worth noting that female prawns require higher levels of dietary folic acid compared to males. In males, high dietary folic acid levels resulted in reduced crude protein contents in the body and a decrease in amylase activity in the hepatopancreas. On the other hand, in females, high dietary folic acid levels did not influence the rude protein content but increased the protease activity in the hepatopancreas. Additionally, as the dietary folic acid levels increased, the sex differentiation in the population was not affected but significantly enhanced fecundity and antioxidant status in both male and female *M. nipponense*. Transcriptome analysis revealed that females responded to thermal stress by regulating pathways associated with “Oxidative phosphorylation” and “Pantothenate and CoA biosynthesis” when fed a high-folic-acid diet. Different from females, male prawns showed a decrease in the expression of metabolism-related genes within the “Fatty acid biosynthesis” signaling pathway. This study provides preliminary insights into the different performances of male and female *M. nipponense* on dietary folic acid. These findings are valuable for improving the overall health and growth of this species in aquaculture production, as well as benefiting other species within the aquaculture industry.

## Figures and Tables

**Figure 1 animals-13-03677-f001:**
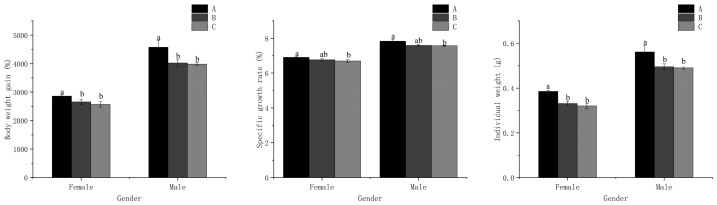
Growth performance for male and female *M. nipponense* fed with experimental diets. (Note: Data were expressed as means ± SD from four replicated groups of prawns (*n* = 4), with 40 prawns in each group. Different superscript letters indicate significant differences (*p* < 0.05)).

**Figure 2 animals-13-03677-f002:**
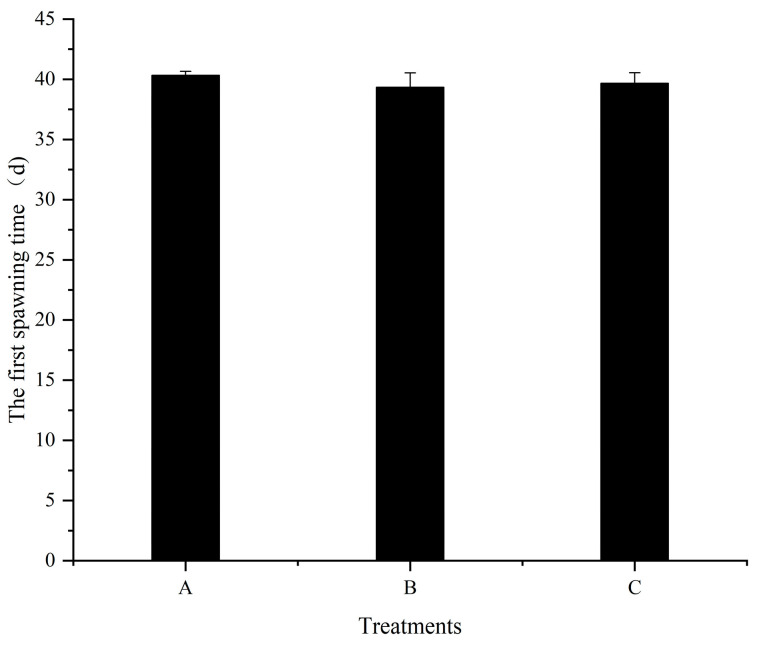
Effects of different dietary folic acid levels on the first spawning time of female *M. nipponense*. (Note: Data were expressed as means ± SD from four replicated groups of prawns (*n* = 4), with 40 prawns in each group).

**Figure 3 animals-13-03677-f003:**
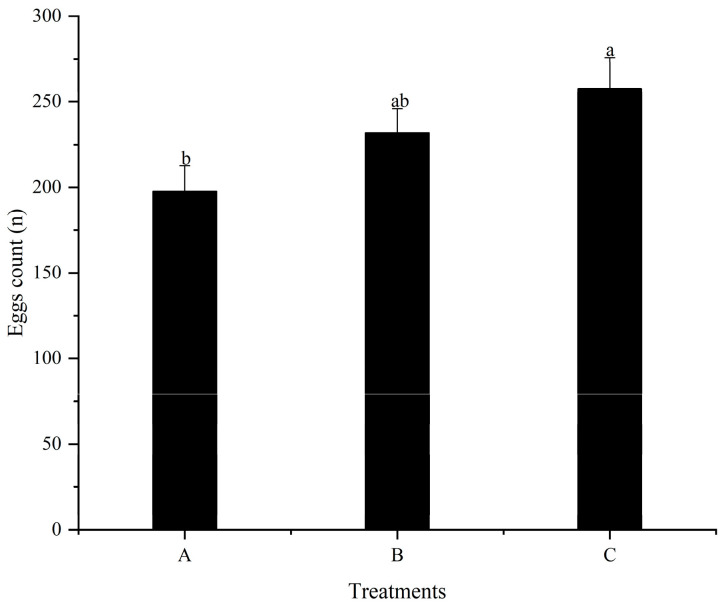
Effects of different dietary folic acid levels on the egg count of female *M. nipponense*. (Note: Data were expressed as means ± SD from four replicated groups of prawns (*n* = 4), with 40 prawns in each group. Different superscript letters indicate significant differences (*p* < 0.05)).

**Figure 4 animals-13-03677-f004:**
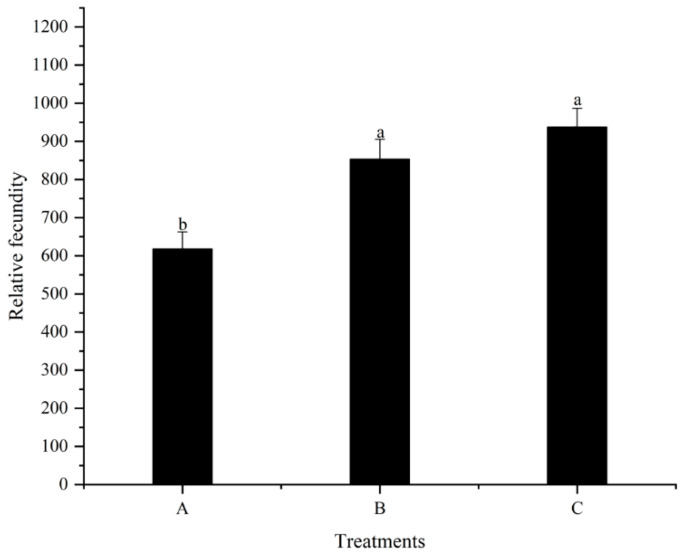
Effects of different dietary folic acid levels on the relative fecundity of female *M. nipponense*. (Note: Data were expressed as means ± SD from four replicated groups of prawns (*n* = 4), with 40 prawns in each group. Different superscript letters indicate significant differences (*p* < 0.05)).

**Figure 5 animals-13-03677-f005:**
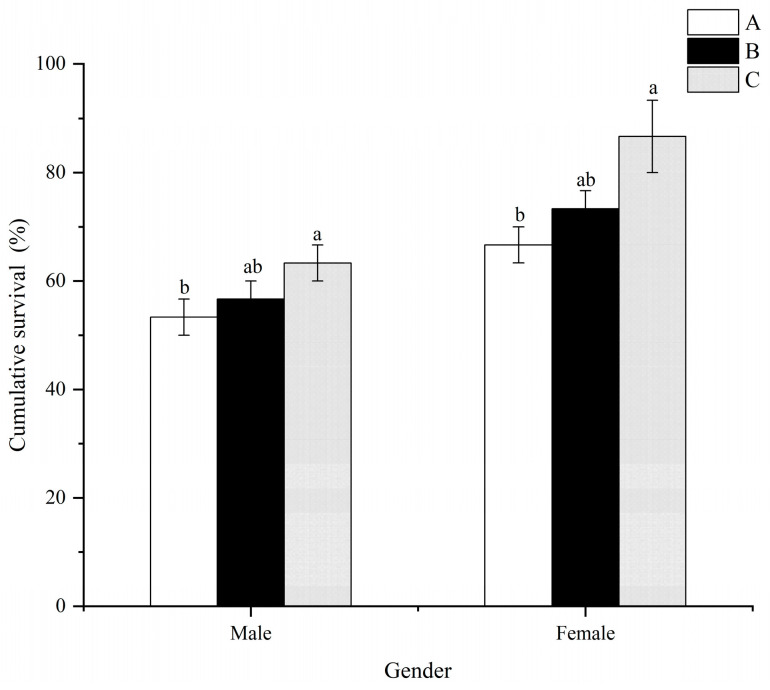
Effects of dietary folic acid levels on the resistance against thermal stress in *M. nipponense. (*Note: Data were expressed as means ± SD from triplicate groups of prawns (*n* = 3), with 10 female and 10 male prawns in each group. Different superscript letters indicate significant differences (*p* < 0.05)).

**Figure 6 animals-13-03677-f006:**
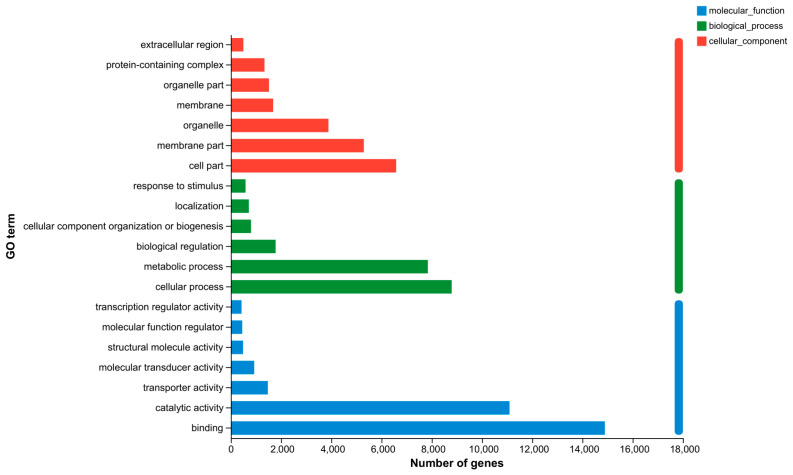
GO classification of unigenes. (Note: The ordinate is the second-level term under the three categories of GO. The abscissa represents the number of genes annotated with the term).

**Figure 7 animals-13-03677-f007:**
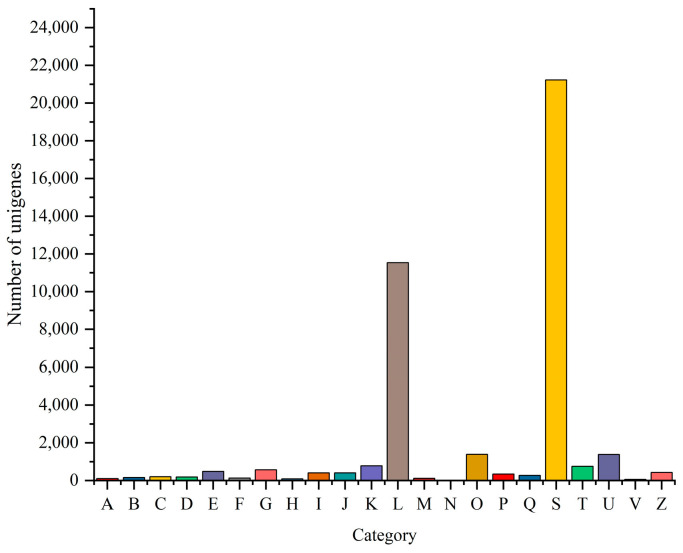
COG function classification of unigenes. (Note: The abscissa is the classification content of COG, and the ordinate is the number of unigenes. A, RNA processing and modification; B, chromatin structure and dynamics; C, energy production and conversion; D, cell cycle control, cell division, and chromosome partitioning; E, amino acid transport and metabolism; F, nucleotide transport and metabolism; G, carbohydrate transport and metabolism; H, coenzyme transport and metabolism; I, lipid transport and metabolism; J, translation, ribosomal structure, and biogenesis; K, transcription; L, replication, recombination, and repair; M, cell wall/membrane/envelope biogenesis; N, cell motility; O, post-translational modification, protein turnover, and chaperones; P, inorganic ion transport and metabolism; Q, secondary metabolites biosynthesis, transport, and catabolism; S, function unknown; T, signal transduction mechanisms; U, intracellular trafficking, secretion, and vesicular transport; V, defense mechanisms; and Z, cytoskeleton).

**Figure 8 animals-13-03677-f008:**
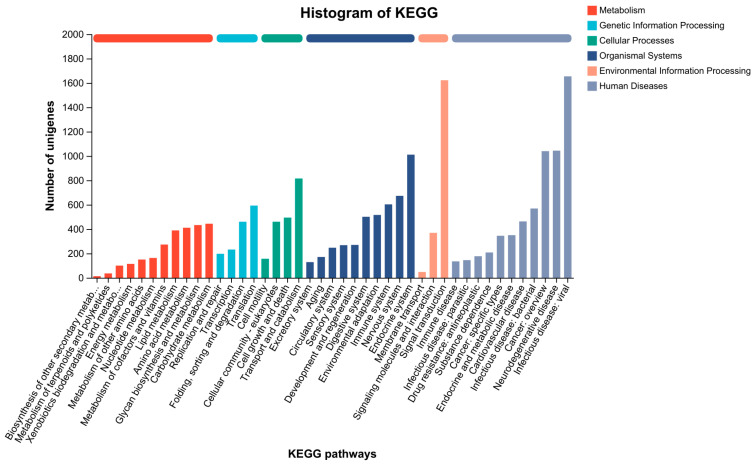
KEGG pathway distribution of unigenes. (Note: The ordinate is the number of unigenes annotated to the pathway. The abscissa is the name of the KEGG signal pathway).

**Figure 9 animals-13-03677-f009:**
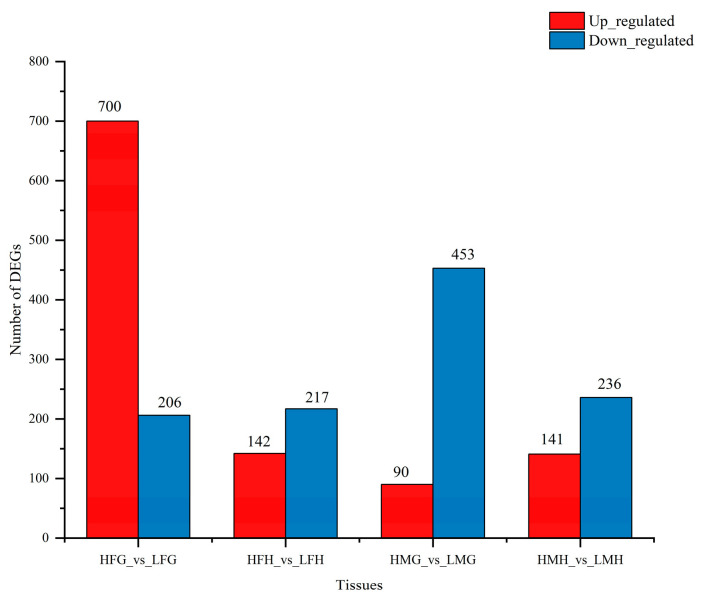
The number of differentially expressed genes in the four tested tissues from male and female prawn *M. nipponense* fed low and high folic acid levels diets. (Note: The upregulated and downregulated DEGs are shown in red and blue, respectively. The abscissa presents the tissues; the ordinate shows the number of DEGs. HFG, the gills tissues of females from treatment C; LFG, the gills tissues of females from treatment A; HFH, the hepatopancreas tissues of females from treatment C; LFH, the hepatopancreas tissues of females from treatment A; HMH, the hepatopancreas tissues of males from treatment C; LMH, the hepatopancreas tissues of males from treatment A; HMG, the gills tissues of males from treatment C; LMG, the gills tissues of males from treatment A).

**Figure 10 animals-13-03677-f010:**
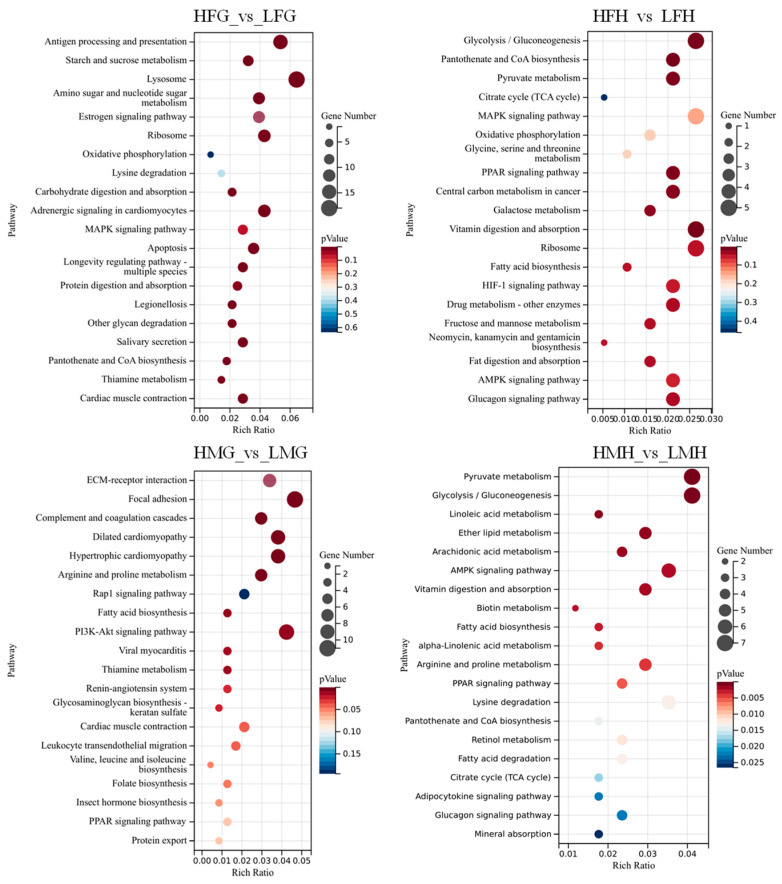
DEGs in KEGG pathway enrichment from the gills and hepatopancreas between male and female prawns fed low and high folic acid levels diets. (Note: The ordinate (**left**) is the name of the KEGG signal pathway, the ordinate (**right**) is the number of unigenes annotated to the pathway and the (*p*) value, and the abscissa is the rich ratio. HFG, the gill tissues of females from treatment C; LFG, the gill tissues of females from treatment A; HFH, the hepatopancreas tissues of females from treatment C; LFH, the hepatopancreas tissues of females from treatment A; HMH, the hepatopancreas tissues of males from treatment C; LMH, the hepatopancreas tissues of males from treatment A; HMG, the gill tissues of males from treatment C; LMG, the gill tissues of males from treatment A).

**Figure 11 animals-13-03677-f011:**
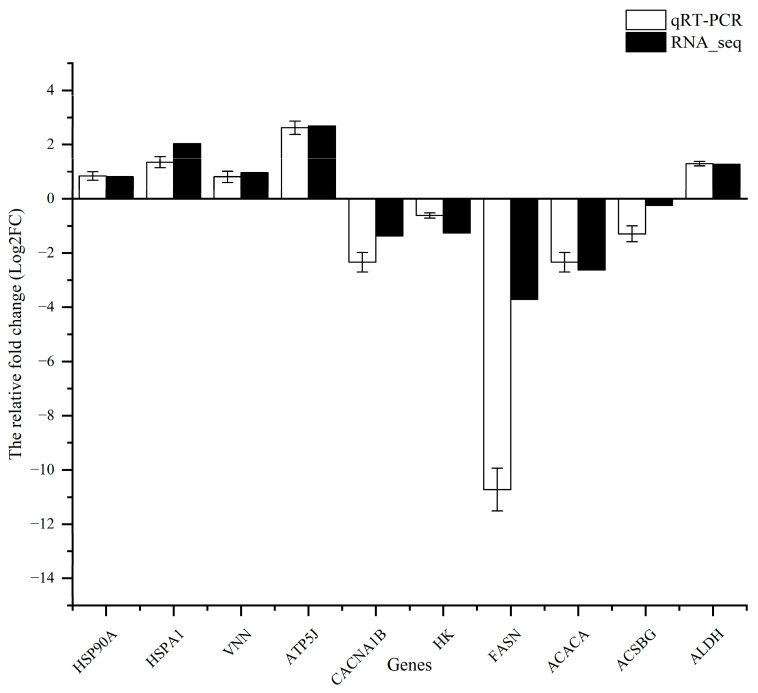
Result comparison of the qRT-PCR and RNA-seq. (Note: the qRT-PCR and RNA-seq results are shown in white and black, respectively. The abscissa presents the name of DEGs; the ordinate shows the expression levels).

**Table 1 animals-13-03677-t001:** Composition of the basic experimental diets (% dry matter basis).

Ingredients (%)	Contents
Flour ^a^	20.9
Soybean meal ^a^	19
Brown fish meal ^a^	19
α-starch ^a^	11.9
Peanut meal ^a^	7.6
Chicken meal ^a^	3.8
Corn protein powder ^a^	3.8
Spray dried blood powder ^a^	2.8
Beer yeast cell ^a^	2.0
Monocalcium phosphate ^a^	2
Squid paste ^a^	2
Fish oil ^a^	1.5
Mineral premix ^b^	1.5
Vitamin premix ^c^	1
Cellulose ^b^	0.5
Soybean lecithin ^a^	0.4
Choline chloride ^a^	0.3
Total	100
Proximate compositions (%)	
Crude protein	39.04
Crude lipid	6.37
Ash	9.35
Moisture	10.24

Note: ^a^ Guangdong Yuehai Feed Co., Ltd. (Zhanjiang, China). ^b^ Per kg mineral premix contains 100 mg of Co; 1400 mg of Cu; 900 mg of Zn; and 450 mg of Mn. ^c^ Per kg vitamin premix contains 1,500,000 IU of vitamin A; 5000 mg of vitamin B 1; 2500 mg of vitamin B; 190,000 IU of vitamin D 3; 180,000,000 IU of vitamin E; 8000 mg of vitamin B 6; 2000 mg of vitamin K 3; 2600 mg of Niacin; 2000 mg of pantothenic acid; and 250 mg of folic acid.

**Table 2 animals-13-03677-t002:** Survival and sex ratio of *M. nipponense* fed with experimental diets.

Parameters	Treatments			ANOVA *p*-Value
	A (1.22 mg/kg)	B (5.44 mg/kg)	C (10.09 mg/kg)	
SR (%)	96.25 ± 2.165	93.75 ± 3.608	92.50 ± 1.767	0.609
Female/male ratio	2.30 ± 0.229	2.15 ± 0.134	2.17 ± 0.225	0.798

Note: Data were expressed as means ± SD from four replicated groups (*n* = 4). Abbreviations: SR, survival rate.

**Table 3 animals-13-03677-t003:** Biochemical analysis (% dry weight basis) in male and female *M. nipponense* bodies fed with experimental diets.

Parameters		Treatments			ANOVA *p*-Value
		A (1.22 mg/kg)	B (5.44 mg/kg)	C (10.09 mg/kg)	
Male	Moisture	74.02 ± 0.057	74.04 ± 0.165	75.49 ± 1.564	0.356
	Ash	15.71 ± 0.870	16.23 ± 0.794	16.61 ± 0.144	0.661
	Crude protein	67.11 ± 0.754 ^a^	64.21 ± 0.497 ^b^	64.53 ± 0.666 ^b^	0.037
	Total lipid	10.40 ± 0.180 ^a^	9.63 ± 0.385 ^a^	8.69 ± 0.633 ^b^	0.023
Female	Moisture	73.27 ± 0.130	73.90 ± 0.465	73.94 ± 0.793	0.566
	Ash	16.48 ± 0.487	15.33 ± 1.053	16.05 ± 0.748	0.612
	Crude protein	65.28 ± 1.484	62.79 ± 1.057	62.91 ± 1.262	0.364
	Total lipid	11.67 ± 0.534 ^a^	10.13 ± 1.892 ^b^	10.78 ± 0.187 ^b^	0.045

Note: Data were expressed as means ± SD from four replicated groups of prawns (*n* = 4), with 40 prawns in each group. Different superscript letters indicate significant differences (*p* < 0.05).

**Table 4 animals-13-03677-t004:** Digestive enzyme activities of male and female *M. nipponese* fed with experimental diets (U/mg protein).

Parameters		Treatments			ANOVA *p*-Value
		A (1.22 mg/kg)	B (5.44 mg/kg)	C (10.09 mg/kg)	
	Amylase	0.96 ± 0.015 ^a^	0.69 ± 0.045 ^b^	0.52 ± 0.013 ^c^	<0.001
Male	Lipase	327.49 ± 5.663 ^a^	284.83 ± 2.414 ^b^	291.79 ± 2.702 ^b^	0.001
	Protease	27.45 ± 1.068 ^c^	45.66 ± 1.627 ^a^	32.53 ± 0.204 ^b^	<0.001
	Amylase	0.54 ± 0.011	0.48 ± 0.028	0.54 ± 0.010	0.079
Female	Lipase	269.42 ± 4.453 ^a^	244.88 ± 2.247 ^b^	235.28 ± 2.241 ^b^	0.001
	Protease	22.84 ^c^ ± 1.428 ^c^	26.98 ± 0.165 ^b^	33.54 ± 1.010 ^a^	0.001

Note: Data were expressed as means ± SD from four replicated groups of prawns (*n* = 4), with 40 prawns in each group. Different superscript letters indicate significant differences (*p* < 0.05).

**Table 5 animals-13-03677-t005:** Effects of dietary folic acid levels on the basal composition of *M. nipponense* eggs (mg/g dry weight basis).

	Treatments			ANOVA *p*-Value
Parameters	A (1.22 mg/kg)	B (5.44 mg/kg)	C (10.09 mg/kg)	
Moisture	722.46 ± 2.766	728.39 ± 9.229	713.77 ± 3.691	0.292
Total protein	35.81 ± 0.316 ^c^	36.86 ± 0.128 ^b^	38.69 ± 0.089 ^a^	<0.001
Total lipid	51.19 ± 0.597 ^c^	53.66 ± 0.970 ^b^	56.48 ± 0.500 ^a^	0.006

Note: Data were expressed as means ± SD from four replicated groups of prawns (*n* = 4), with 40 prawns in each group. Different superscript letters indicate significant differences (*p* < 0.05).

**Table 6 animals-13-03677-t006:** Effects of dietary folic acid levels on antioxidant capacities after the resistance against thermal stress in *M. nipponense.*

Parameters	Treatments				ANOVA *p*-Value
		A (1.22 mg/kg)	B (5.44 mg/kg)	C (10.09 mg/kg)	
Male	T-AOC (U/mL)	0.32 ± 0.010 ^b^	0.28 ± 0.012 ^c^	0.37 ± 0.011 ^a^	<0.001
	SOD (U/mL)	98.63 ± 2.98 ^b^	106.89 ± 0.052 ^a^	107.27 ± 0.333 ^a^	0.02
	CAT (U/mL)	5.56 ± 0.241 ^b^	6.90 ± 0.526 ^a^	6.42 ± 0.207 ^ab^	0.045
	MDA (mmol/mL)	152.81 ± 3.510 ^a^	148.07 ± 1.384 ^b^	126.38 ± 0.642 ^c^	<0.001
Female	T-AOC (U/mL)	0.24 ± 0.013 ^c^	0.32 ± 0.014 ^b^	0.40 ± 0.012 ^a^	<0.001
	SOD (U/mL)	111.31 ± 2.16 ^b^	121.21 ± 1.40 ^a^	117.14 ± 0.882 ^a^	0.012
	CAT (U/mL)	7.10 ± 0.193	7.74 ± 0.285	7.31 ± 0.316	0.307
	MDA (mmol/mL)	134.91 ± 4.577 ^a^	131.80 ± 0.936 ^a^	117.31 ± 3.187 ^b^	0.019

Note: Data were expressed as means ± SD from triplicate groups of prawns (*n* = 3), with 10 female and 10 male prawns in each group. Different superscript letters indicate significant differences (*p* < 0.05). T-AOC, total antioxidant capacity; SOD, superoxide dismutase; CAT, catalase; and MDA, malondialdehyde.

**Table 7 animals-13-03677-t007:** Summary of the annotation statistics of the unigenes of the transcriptome of male and female *M. nipponense* prawns fed with the low and high folic acid supplementation diets.

Database	Number of Annotated Unigenes	%
Annotated in NR	39,753 (0.922)	92.20
Annotated in SwissProt	23,772 (0.5514)	55.14
Annotated in KEGG	13,145 (0.3049)	30.49
Annotated in COG	33,514 (0.7773)	77.73
Annotated in PFAM	28,185 (0.6537)	65.37
Annotated in GO	23,093 (0.5356)	53.56
Total annotation	40,085 (0.9297)	92.97
Total unigenes	43,115	100

**Table 8 animals-13-03677-t008:** The genes with a few significantly changed KEGG pathways after thermal stress.

DEGs	Annotation	Tissues	Log2Foldchange
CTSA	cathepsin A (carboxypeptidase C)	HFG_vs._LFG	3.11
GLB1	beta-galactosidase	HFG_vs._LFG	3.68
NPC2	Niemann-Pick C2 protein	HFG_vs._LFG	6.22
HSP90A	molecular chaperone HtpG	HFG_vs._LFG	0.81
HSPA1	heat shock 70 kDa protein	HFG_vs._LFG	2.03
ALDO	fructose-bisphosphate aldolase, class I	HFH_vs._LFH	−0.48
PGAM	2,3-bisphosphoglycerate-dependent phosphoglycerate mutase	HFH_vs._LFH	−0.47
HK	Hexokinase	HFH_vs._LFH	−1.26
ENO	Enolase	HFH_vs._LFH	−0.46
CACNA1B	voltage-dependent calcium channel N type alpha-1B	HFH_vs._LFH	−1.37
HSPA1/6/8	heat shock 70 kDa protein 1/6/8	HFH_vs._LFH	3.17
VEGFB	vascular endothelial growth factor B	HMH_vs._LMH	−0.52
LAMC1	laminin, gamma 1	HMH_vs._LMH	−3.15
VEGFB	vascular endothelial growth factor B	HMH_vs._LMH	−0.52
ITGA8	integrin alpha 8	HMH_vs._LMH	0.03
ACTB/G1	integrin beta 5 (ITGB5) actin beta/gamma 1	HMH_vs._LMH	−1.3
RAC1	Ras-related C3 botulinum toxin substrate 1	HMH_vs._LMH	0.12
ATP5J	F-type H+-transporting ATPase subunit 6	HFG_vs._LFG	2.68
ALDH	aldehyde dehydrogenase (NAD+)	HFG_vs._LFG	1.28
VNN	pantetheine hydrolase	HFG_vs._LFG	5.97
FASN	fatty acid synthase, animal type	HMG_vs._LMG	−3.71
ACACA	acetyl-CoA carboxylase/biotin carboxylase 1	HMG_vs._LMG	−2.63
ACSBG	long-chain-fatty-acid-CoA ligase ACSBG	HMG_vs._LMG	−0.25

Note: HFG, the gill tissues of females from treatment C; LFG, the gill tissues of females from treatment A; HFH, the hepatopancreas tissues of females from treatment C; LFH, the hepatopancreas tissues of females from treatment A; HMH, the hepatopancreas tissues of males from treatment C; LMH, the hepatopancreas tissues of males from treatment A; HMG, the gill tissues of males from treatment C; LMG, the gill tissues of males from treatment A.

## Data Availability

The data presented in this study are available from the corresponding authors upon reasonable request.

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
