# Peer review of "Effects of Dietary Folic Acid Supplementation on Sex Differences in Oriental River Prawn, *Macrobrachium nipponense"

_animals, 2023, doi:10.3390/ani13233677_

Round 1

Reviewer 1 Report

Comments and Suggestions for Authors

The manuscript: Effects of dietary folic acid supplementation on sex differences in oriental river prawn, Macrobrachium nipponense. Unfortunately, I should reject this manuscript due to the following:

1.    There is an issue in the manuscript conceptualization and the methodology. English style and grammar need to improve.

2.                No novelty.  

3.                Low quality of presentation, low quality of figures, and several mistakes have been observed.

4.                Lines 82-85: Something wrong with the concentrations and doses. Please, revise the content, regarding 0, 4, and 8 mg/kg.

5.        Line 89: “90 for 20 min This temperature damages all diet contents of Vitamins! Why authors did not use a sun dryer or oven at low temperatures with long 24 or 48 h??

6.                Lines 511-512: “This present finding demonstrated that female prawns require higher levels of dietary folic acid compared to males”, this is a basic requirement in any females comparing to any males! regarding the nature of the female's body, what is the new in the conclusion?

Comments on the Quality of English Language

Extensive editing of English language required

Author Response

Dear reviewer,

We would like to express our sincere gratitude for the time and effort you dedicated to reviewing the previous version of our manuscript. Your valuable suggestions have significantly contributed to the improvement of our work. We have carefully considered and incorporated your constructive comments and suggestions into the revised manuscript. Based on the instructions provided in your letter, we have uploaded the file of the revised manuscript. Accordingly, we have uploaded a copy of the original manuscript with all changes highlighted by using the track changes mode in MS Word. In the even that we missed any one of the comments please let me know.

In the following section, we have provided a detailed point-by-point response to address each of your comments and suggestions (highlighted in red). We aim to ensure that this revised manuscript adequately addresses all of your concerns. We greatly appreciate your diligent assessment and sincerely hope that the revisions made meet with your approval.

Lastly, we revised the whole manuscript carefully to avoid language errors. We believe that the language is now acceptable for the review process. Once again, thank you very much for your comments and suggestions. Because of this, our manuscripts will become more scientifically accurate, structurally complete, and logical.

Sincerely,

  • There is an issue in the manuscript conceptualization and the methodology. English style and grammar need to improve.

RESPONSE: Thank you very much for your valuable suggestions. For issues of the English style and grammar in the manuscript, we checked the whole manuscript carefully to avoid language errors. Furthermore, we do invite a friend of ours who is a native English speaker from the USA to help polish our article. We hope the revised manuscript will be acceptable to you.

  • No novelty.

RESPONSE: Thank you very much for the positive comments, which are highly appreciated. Similar to Macrobrachium rosenbergii, Macrobrachium nipponense also exhibits a sexually dimorphic growth pattern, with male prawns typically growing faster and reaching larger sizes at harvest time. Based on this background, it raises questions about potential differences in folic acid requirements, metabolic capacity, and reproduction between male and female M. nipponense, as well as potential gender-specific effects. Therefore, the purpose of this experiment is to further enhance the economic and social benefits associated with this species.

From the results of the current study, we observed that the male prawns exhibited a higher tolerance to thermal stress than the male prawns when fed with the same folic acid level. Additionally, through transcriptome sequencing, we aimed to explore the underlying molecular mechanisms responsible for these observed differences. We acknowledge the value of your suggestions in refining our experimental design for future studies. Nevertheless, we believe that the results obtained from this experiment are significant not only for M. nipponense but also for species exhibiting sexual dimorphic growth patterns. 

  • Low quality of presentation, low quality of figures, and several mistakes have been observed.

RESPONSE: We are grateful for the suggestion. Firstly, we are very sorry for the difficulty caused to your reading because of our carelessness, especially in the details. Based on this, we checked the whole manuscript, and a lot of changes have been made in our revised manuscript. We sincerely hope that the revised manuscript will be acceptable to you.

  • Lines 82-85: Something wrong with the concentrations and doses. Please, revise the content, regarding 0, 4, and 8 mg/kg.

RESPONSE: Thank you for your kind reminder, we have revised the content for the folic acid.

  • Line 89: “90℃ for 20 min” This temperature damages all diet contents of Vitamins! Why authors did not use a sun dryer or oven at low temperatures with long 24 or 48 h?

RESPONSE: We feel great thanks for your professional review work on our article. Your comments are really thoughtful. As you said, high temperatures will damage all diet contents of Vitamins, and use a sun dryer or oven at low temperatures with long 24-48 hours is the most appropriate and scientific way. But the fact is that we dry the raw diet indoors. Furthermore, considering the effects of light on dietary folic acid contents, we stored the prepared diet in brown ziplock bags in the refrigerator at 4℃ for a long time and only took out an appropriate amount of feed every day. Finally, according to the final measured value, we found that the dietary folic acid levels in each treatment were within a reasonable range.

However, the reason why we wrote “90℃ for 20 min” in the manuscript is that our research group has prepared various feeds for a long time, so we did not change the content of this part due to carelessness in referring to the published article on diet preparation. We are very sorry for our carelessness and awkwardness, thank you once again for your valuable contribution to our study.

  • Lines 511-512: “This present finding demonstrated that female prawns require higher levels of dietary folic acid compared to males”, this is a basic requirement in any females comparing to any males! regarding the nature of the female's body, what is the new in the conclusion?

RESPONSE: Thank you very much for the positive comments, which are highly appreciated. In our study, we conducted a comparison of growth performance, reproductive performance, and tolerance to high temperatures between male and female prawns. Our findings indicate that female prawns require higher levels of dietary folic acid compared to males. Additionally, through transcriptome analysis, we identified notable differences in the intrinsic regulatory processes between females and males.

Specifically, when fed with a high folic acid diet, females responded to thermal stress by regulating genes associated with "Oxidative phosphorylation" and "Pantothenate and CoA bio-synthesis" signaling pathways. In contrast, male prawns exhibited a decrease in the expression of metabolism-related genes within the "Fatty acid biosynthesis" signaling pathway. These results provide insights into the physiological perspectives of both male and female prawns, which can contribute to the development of scientific, reasonable, and standardized breeding practices for this species

We would like to express our sincere gratitude to you for providing invaluable suggestions. Thank you once again for your valuable contribution to our study.

Reviewer 2 Report

Comments and Suggestions for Authors

This paper is among the most readable papers I have ever read. I definitely enjoy seeing studies on Macrobrachium prawns. I have few suggestions for consideration:

General: Is there a particular source of folic acid that is attractive to prawns or is it ubiquitous in aquatic ecosystems? 

Lines 46 – 50: It was kind of strange reading the insect examples. Could you find an example of an invertebrate or shrimp/ prawn that is used for aquaculture? If not, I would just state that in general differences between male and female prawns are unknown, and generalize/ reduce the insect example. I see this in the last phrase of the paragraph for M nipponense, so make the prior phrases more generalized, i.e. in crustaceans.

Line 71: change seeds to post-larvae

Line 73: instead of using juveniles, use prawns because you will have previously stated that they are post-larvae.

Line 83: describe treatments as treatments A, B, and C because they were listed further in the text and I had no idea what they were.

Line 104: This paragraph was difficult to read.

Lines 431 and 434: Again, I don’t know that it is a great idea to compare rats and Drosophila with freshwater prawns. Perhaps find more relatable examples if possible?

Line 472: delete potential

Author Response

Dear reviewer,

we extend our most sincere thanks to you for the time and effort that you have put into reviewing the previous version of the manuscript. Those suggestions have enabled us to improve our work. We have tried our best to revise the manuscript according to your kind and construction comments and suggestions. Based on the instructions provided in your letter, we have uploaded the file of the revised manuscript. Accordingly, we have uploaded a copy of the original manuscript with all changes highlighted by using the track changes mode in MS Word. In the even that we missed any one of the comments please let me know.

We have provided a point-by-point response to your comments below in red color. Please find the following detailed responses to your comments and suggestions. We sincerely hope that this revised manuscript has addressed all your comments and suggestions. We appreciate your warm work earnestly and hope that the correction will meet with approval.

Lastly, we revised the whole manuscript carefully to avoid language errors. We believe that the language is now acceptable for the review process. Once again, thank you very much for your comments and suggestions. Because of this, our manuscripts will become more scientifically accurate, structurally complete, and logical. We hope that the revised manuscript will meet with approval. If any further action is needed, please let us know immediately. We look forward to hearing back from you.

Sincerely

  • General: Is there a particular source of folic acid that is attractive to prawns or is it ubiquitous in aquatic ecosystems?

RESPONSE: Thank you for your kind question. Folic acid (FA) is an essential nutrient, specifically a water-soluble form of vitamin B9. It plays a crucial role in modulating the transfer of one-carbon donor or acceptor units for protein and DNA synthesis, methylation, and gene expression. While there is currently no information available on whether folic acid is attractive to prawns or present ubiquitously in aquatic ecosystems, it does play an important role in amino acid and nucleotide metabolism, as well as the growth and health of most aquatic animals. This nutrient has been extensively studied in various aquatic animals, such as Macrobrachium rosenbergii, Eriocheir sinensis, Megalobrama amblycephala, and others.

Similar to M. rosenbergii, Macrobrachium nipponense exhibits a sexually dimorphic growth pattern, with male prawns typically growing faster and reaching larger sizes at harvest time. Given this background, it raises questions about potential differences in folic acid requirements, metabolic capacity, and reproduction between male and female M. nipponense, as well as potential gender-specific effects. Therefore, this experiment is designed to further enhance the economic and social benefits associated with this species.

Reference:

Annamalai A, Saravana Bhavan P, Karuppaiya V: Effects of different levels of dietary folic acid on the growth performance, muscle composition, immune response, and antioxidant capacity of freshwater prawn, Macrobrachium rosenbergii. Aquaculture 2016, 464:136-144

Wei JJ, Zhang F, Tian WJ, Kong YQ, Li Q, Yu N, Du ZY, Wu QQ, Qin JG, Chen LQ: Effects of dietary folic acid on growth, antioxidant capacity, non-specific immune response, and disease resistance of juvenile Chinese mitten crab Eriocheir sinensis (Milne-Edwards, 1853). Aquaculture Nutrition 2016, 22(3):567-574.

Sesay DF-T, Habte-MichaelZhou, QunlanRen, MingchunXie, JunLiu, BoChen, RuliPan, Liangkun: Effects of dietary folic acid on the growth, digestive enzyme activity, immune response and antioxidant enzyme activity of blunt snout bream (Mega-lobrama amblycephala) fingerling. Aquaculture 2016, 452(Null).

  • Lines 46 – 50: It was kind of strange reading the insect examples. Could you find an example of an invertebrate or shrimp/ prawn that is used for aquaculture? If not, I would just state that in general differences between male and female prawns are unknown, and generalize/ reduce the insect example. I see this in the last phrase of the paragraph for M nipponense, so make the prior phrases more generalized, i.e. in crustaceans.

RESPONSE: We sincerely appreciate the valuable suggestions. We have searched the literature carefully but were unable to find an example of an invertebrate or shrimp/prawn. However, we have incorporated additional fish samples, including Sardinops sagax, Euphylax dovii, and Stenocionops ovata, into the INTRODUCTION part of our revised manuscript.

  • Line 71: change seeds to post-larvae.
  • Line 73: instead of using juveniles, use prawns because you will have previously stated that they are post-larvae.
  • Line 472: delete potential

RESPONSE: For the above three suggestions, we sincerely thank you for your careful reading. As suggested, we have corrected the “seeds” into “post-larvae”, and “juveniles” into “prawns” and deleted “potential”.

  • Line 83: describe treatments as treatments A, B, and C because they were listed further in the text and I had no idea what they were.

RESPONSE: We feel very sorry for our carelessness. In our resubmitted manuscript, the treatments A, B, and C have been described. Thanks for your kind reminder.

  • Line 104: This paragraph was difficult to read.

RESPONSE: We are very sorry for the difficulty caused to your reading because we did not describe this paragraph in detail. Therefore, we have rewritten this paragraph.

  • Lines 431 and 434: Again, I don’t know that it is a great idea to compare rats and Drosophila with freshwater prawns. Perhaps find more relatable examples if possible?

RESPONSE: We feel very sorry that we gave inappropriate examples, so according to your valuable suggestions, we have revised the references and given a fish example (Ayu Plecoglossus altivelis).

Lastly, we revised the whole manuscript carefully to avoid language errors. We believe that the language is now acceptable for the review process. Once again, thank you very much for your comments and suggestions.

Reviewer 3 Report

Comments and Suggestions for Authors

Abstract:

The abstract should follow the sequence used in the manuscript. E.g., Growth performances should be mentioned before hepatopancreas protease activity. Moreover, the aim of the project should be clearly stated.

Line 23: Dietary lipid level was not increasing. What do you mean with that?

Introduction:

It is too short and do not clearly describes the state of art of the folic acid feed inclusion in prawn nutrition. Moreover, the aim of the study should be described more clearly and concisely, leaving analytical details for the Material and Methods section.

Material and methods

Line 70-73 – The count of individuals, tanks, replicates and treatment seems not correct. In fact:

·   480 individuals / 12 tanks = 40 individuals per tank

·   3 treatments x 3 replicates = 9 tanks rather then 12

What wrong in the described experimental set up? Please, check it out carefully

Line 83 - Please, refer folic acid source (commercial name and Company) and its main characteristics, such as folic acid concentration

Line 92 (table 1) - The components should be listed according to their content, in a decrescent order

Line 92 (table 1 – total) - Carbohydrate (NSC and SC) content is missing. Please provide them in order to reach 100% of the feed composition

Line 92 (table 1 – note) - Since the exact composition is known, please, put each ingredient of the protein mixture in the table together to the other ingredients. Moreover, the content of each mineral and vitamins have to be precisely provided. It is not correct providing so wide ranges.

Line 108 - 1) adjust the font size and 2) what do you mean with “anatomic microscope”? Please, describe more properly the whole procedure used for sex determination

Line 118 - Rather than “measured”, it would be better writing “analysed according to”

Line 120 - May you provide model and company of the muffel?

Line 123 – write “body samples” rather than “whole-body”

Line 125 – Prawn were kept in “tank”, not in cage (see above description)

Line 138 - This sentence is not clear. Please, write more clearly

Line 170-171 - 1) SD values should be given with one more significant decimal digit than its mean. E.g., if it is used 2 significant decimals for mean values, than 3 significant decimal sholuld be used for SD. Please check it out all the table in the manuscript and revise them accordingly

2) Did you detect not normally distributed data? What did you do in that case? Please describe the used method more exhaustively

Line 183 - This sentence may be redundant. Just mention the table or graph, between brackets, in the text. Similar cases are present throughout the manuscript. Please, consider revising all the manuscritpt

Results

Line 184-189 - The decresing trend is not linear and not all the statistically significant differences are properly described. E.g., for BWG there is no mention of the difference between A and B, for both females and males. Similarly,  the same differences are not mentioned for Individual weight. To this regard, in the graph the heading of the Y axis should modified into “individual weight” (rather than “Weight for each prawn”)

Line 198-200 - A lower fat content is detected for females also between A and B. Please, describe properly all the statistically significant differences

Line 211-212 - 1) in the table, the first described enzyme is Amylase, then, this is the first to be described in the text, before Lipase. 2) Write “lipase activity values” rather than “lipase values” 3) Lipase activities did not decrease as described in the text. E.g., for male it is higher for B in comparison to C, and both of them are significantly different from A (not only C)

Line 226–228 - 1) What do you mean with “Based on the consistent value”? 2) The Fig. 2 may be deleted since it does not provide relevant information.

Line 229-230 - Please, describe differences among groups properly and in detail

Line 234-236 - There are differences also between B and A. Please, mention also these differences

Line 255 onwards - Please, based on the above comments, check-out also the other sections of the results and improve statistical differences description. In doing that, make you sure all the differences will be described in a proper manner.

Line 399-401 - Please, write only text that is relevant for the readers and that provide significant informations

Line 402 - Just “growth” not “optimal growth”. This latter cannot be improved otherwise it would not be “optimal”

Line 404 - What do you mean with “liver saturation”?

Line 449 - What do you mean with “egg nutrition”?

Conclusions

Conclusions may be improved by describing the impact of these findings, e.g., on the aquaculture industry, if any.

Line 511 - Maybe is more prudent and appropriate writing “suggests” rather than “demonstrated”. E.g., regarding growth, females do not require higher dietary folic acids

Line 512-513 - To this regards, these are not the only differences observed. Why do you think these are more worthy to be mentioned than others?

Comments on the Quality of English Language

Regarding the English language, there are not severe (some are present) grammar error. For a contrary, slight improvements might be done under the scientific English language perspectives.

Howevere, in general the English language of the manuscript can be considered more the acceptable.

Author Response

Dear reviewer,

We would like to express our sincere gratitude for the time and effort you dedicated to reviewing the previous version of our manuscript. Your valuable suggestions have significantly contributed to the improvement of our work. We have carefully considered and incorporated your constructive comments and suggestions into the revised manuscript. Based on the instructions provided in your letter, we have uploaded the file of the revised manuscript. Accordingly, we have uploaded a copy of the original manuscript with all changes highlighted by using the track changes mode in MS Word. In the even that we missed any one of the comments please let me know.

In the following section, we have provided a detailed point-by-point response to address each of your comments and suggestions (highlighted in red). We aim to ensure that this revised manuscript adequately addresses all of your concerns. We greatly appreciate your diligent assessment and sincerely hope that the revisions made meet with your approval.

Lastly, we revised the whole manuscript carefully to avoid language errors. We believe that the language is now acceptable for the review process. Once again, thank you very much for your comments and suggestions. Because of this, our manuscripts will become more scientifically accurate, structurally complete, and logical.

Sincerely

  • The abstract should follow the sequence used in the manuscript. E.g., Growth performances should be mentioned before hepatopancreas protease activity. Moreover, the aim of the project should be clearly stated.

RESPONSE: Thank you very much for the positive comments, which are highly appreciated. Based on your valuable suggestions, we have revised the sequence of growth performances and stated the aim of this project clearly in the ABSTRACT section.

  • Line 23: Dietary lipid level was not increasing. What do you mean with that?
  • Line 118 - Rather than “measured”, it would be better writing “analysed according to”.
  • Line 123 – write “body samples” rather than “whole-body”.
  • Line 125 – Prawn were kept in “tank”, not in cage (see above description).
  • Line 402 - Just “growth” not “optimal growth”. This latter cannot be improved otherwise it would not be “optimal”
  • Line 404 - What do you mean with “liver saturation”?
  • Line 511 - Maybe is more prudent and appropriate writing “suggests” rather than “demonstrated”. E.g., regarding growth, females do not require higher dietary folic acids

RESPONSE: For the above seven suggestions, we sincerely thank you for your careful reading. Firstly, we feel very sorry for our carelessness. In our resubmitted manuscript, the typo is revised. Thanks for your kind reminder. In fact. we wanted to use “folic acid” not “lipid”, and we have corrected the “lipid” into “folic acid”. Then, as suggested, we also have collected “measured” into “analyzed”, “whole-body” into “body samples”, “cage” into “tank”, “liver saturation” into “hepatosomatic indexes”, “demonstrated” into “suggests” and delete “optimal”.

  • It is too short and do not clearly describes the state of art of the folic acid feed inclusion in prawn nutrition. Moreover, the aim of the study should be described more clearly and concisely, leaving analytical details for the Material and Methods section.

RESPONSE: We sincerely appreciate the valuable suggestions. As per your kind suggestions, we have described the state of the art of the folic acid feed inclusion in Macrobrachium rosenbergii, Penaeus monodon, Fenneropenaeus chinensis, and Eriocheir sinensis. Additionally, we also added the quantitative requirements of folic acid in these species. Also, we described the aim of the study more clearly and concisely.

  • Line 70-73 – The count of individuals, tanks, replicates and treatment seems not correct. In fact:

480 individuals / 12 tanks = 40 individuals per tank

3 treatments x 3 replicates = 9 tanks rather then 12

What wrong in the described experimental set up? Please, check it out carefully.

RESPONSE: We sincerely thank you for your careful reading, but we are very sorry that the content of this part is not described clearly. In the breeding experiment, each treatment had 4 replicates, and each replicate had 40 larvae, so there were 480 individuals in total. The cage we used is in triplicate, which may be the main reason for the misunderstanding of this part.

  • Line 83 - Please, refer folic acid source (commercial name and Company) and its main characteristics, such as folic acid concentration.
  • Line 92 (table 1) - The components should be listed according to their content, in a decrescent order.
  • Line 92 (table 1 – note) - Since the exact composition is known, please, put each ingredient of the protein mixture in the table together to the other ingredients. Moreover, the content of each mineral and vitamins have to be precisely provided. It is not correct providing so wide ranges.
  • Line 120 - May you provide model and company of the muffel.
  • Line 92 (table 1 – note) - Since the exact composition is known, please, put each ingredient of the protein mixture in the table together to the other ingredients. Moreover, the content of each mineral and vitamins have to be precisely provided. It is not correct providing so wide ranges.
  • Line 170-171 - 1) SD values should be given with one more significant decimal digit than its mean. E.g., if it is used 2 significant decimals for mean values, than 3 significant decimal sholuld be used for SD. Please check it out all the table in the manuscript and revise them accordingly.

RESPONSE: Thank you very much for the positive comments, which are highly appreciated. As your valuable suggestions, we have added the relevant information in the revised version.

  • Line 92 (table 1 – total) - Carbohydrate (NSC and SC) content is missing. Please provide them in order to reach 100% of the feed composition.

RESPONSE: We sincerely appreciate the valuable suggestions. Unfortunately, we didn't test the carbohydrate content in the diet. However, this is not the first time in this experiment, and we can find that carbohydrate content has not been detected in many similar articles that have been published, such as Li (2020), Cornel Angela (2020), and Wei (2016) et al. Anyway, we sincerely thank you very much for your suggestions, and we will improve them in the later experiments.

Reference:

Li L , Wang W , Yusuf A ,et al. Effects of dietary lipid levels on the growth, fatty acid profile and fecundity in the oriental river prawn, Macrobrachium nipponense. Aquaculture Research, 2020, 51.

Cornel Angela, Weilong W, Hongyu L, et al. "The effect of dietary supplementation of Astragalus membranaceus and Bupleurum chinense on the growth performance, immune-related enzyme activities and genes expression in white shrimp, Litopenaeus vannamei." Fish & Shellfish Immunology, 2020, 107:379-384.

Wei JJ, Zhang F, Tian WJ, Kong YQ, Li Q, Yu N, Du ZY, Wu QQ, Qin JG, Chen LQ: Effects of dietary folic acid on growth, an-tioxidant capacity, non-specific immune response and disease resistance of juvenile Chinese mitten crab Eriocheir sinensis (Milne-Edwards, 1853). Aquaculture Nutrition 2016, 22(3):567-574.

  • Line 108 - 1) adjust the font size and 2) what do you mean with “anatomic microscope”? Please, describe more properly the whole procedure used for sex determination.

RESPONSE: Thank you for your kind reminder, and we have adjusted the font size. Additionally, we are very sorry for the difficulty caused to your reading because we did not describe this paragraph in detail. Therefore, according to your kind suggestion, we have rewritten this paragraph and described more properly the whole procedure used for the sex distinction.

  • Line 138 - This sentence is not clear. Please, write more clearly.

RESPONSE: We are very sorry for the difficulty caused to your reading because we did not describe this paragraph in detail. Therefore, we have rewritten this paragraph.

  • Did you detect not normally distributed data? What did you do in that case? Please describe the used method more exhaustively

RESPONSE: Thank you very much for your kind reminder, we detected the non-normal distribution. For the data of non-normal distribution, Natural Log transformation was utilized to transfer non-normal distribution variables into normal distribution variables. We have added this description in the revised version.

  • Line 183 - This sentence may be redundant. Just mention the table or graph, between brackets, in the text. Similar cases are present throughout the manuscript. Please, consider revising all the manuscript.

RESPONSE: Thank you very much for the positive comments, which are highly appreciated. As your valuable suggestions, we have revised the whole in the revised version.

  • Line 184-189 - The decresing trend is not linear and not all the statistically significant differences are properly described. E.g., for BWG there is no mention of the difference between A and B, for both females and males. Similarly, the same differences are not mentioned for Individual weight. To this regard, in the graph the heading of the Y axis should modified into “individual weight” (rather than “Weight for each prawn”)
  • Line 198-200 - A lower fat content is detected for females also between A and B. Please, describe properly all the statistically significant differences
  • Line 211-212 - 1) in the table, the first described enzyme is Amylase, then, this is the first to be described in the text, before Lipase. 2) Write “lipase activity values” rather than “lipase values” 3) Lipase activities did not decrease as described in the text. E.g., for male it is higher for B in comparison to C, and both of them are significantly different from A (not only C)
  • Line 226–228 - 1) What do you mean with “Based on the consistent value”? 2) The Fig. 2 may be deleted since it does not provide relevant information
  • Line 229-230 - Please, describe differences among groups properly and in detail
  • Line 234-236 - There are differences also between B and A. Please, mention also these differences
  • Line 255 onwards - Please, based on the above comments, check-out also the other sections of the results and improve statistical differences description. In doing that, make you sure all the differences will be described in a proper manner

RESPONSE: We feel great thanks for your professional review work on our article. Your comments are thoughtful. As you are concerned, several problems need to be addressed. According to your nice suggestions, we have rewritten this RESULT section according to your valuable suggestions. Once again, thank you very much for your comments and suggestions.

  • Line 399-401 - Please, write only text that is relevant for the readers and that provide significant informations.

RESPONSE: We feel very sorry for our carelessness. In our resubmitted manuscript, the text is revised. Thanks for your kind reminder.

  • Line 449 - What do you mean with “egg nutrition”?

RESPONSE: Thank you for your kind question. The nutritional status of broodstock could markedly affect the eggs and offspring quality of decapod crustaceans by transferring nutrients and energy to the eggs from ovaries (Wu et al., 2011). Conversely, the biochemical compositions of eggs also reflect the physiological status of spawners. Therefore, in the current study, we wanted to explore the effects of dietary folic acid on the reproductive performance of M. nipponense by detecting the nutrient level of eggs. Furthermore, the importance of this parameter was also highlighted in the study conducted by Li et al. (2020).

Reference:

Li L, Wang W, Yusuf A, et al. Effects of dietary lipid levels on the growth, fatty acid profile, and fecundity in the oriental river prawn, Macrobrachium nipponense. Aquaculture Research, 2020, 51.

Wu, X G., Wang, Z K, Cheng, Y X., et al. Effects of dietary phospholipids and highly unsaturated fatty acids on the precocity, survival, growth and hepatic lipid composition of juvenile Chinese mitten crab, Eriocheir sinensis (H. Milne-Edwards). Aquaculture Research, 2011, 42, 457–468.

  • Conclusions may be improved by describing the impact of these findings, e.g., on the aquaculture industry, if any.

RESPONSE: We sincerely appreciate the valuable comments. We have revised the conclusion and added a description of the impact of these findings on the aquaculture industry.

  • Line 512-513 - To this regards, these are not the only differences observed. Why do you think these are more worthy to be mentioned than others?

RESPONSE: Thank you very much for the positive comments, which are highly appreciated. We are very sorry for our inappropriate description. Therefore, according to your kind suggestions, we have rewritten this part.

Round 2

Reviewer 1 Report

Comments and Suggestions for Authors

Accept in present form

Author Response

Dear Reviewer,

I hope this email finds you well. I wanted to express my sincere gratitude for your thoughtful and constructive feedback on our manuscript titled " Effects of dietary folic acid supplementation on sex differences in oriental river prawn, Macrobrachium nipponense." It is with great pleasure that I received your positive response, indicating that the paper has been accepted for publication after the first round of revisions.

Your expertise and insightful comments have immensely contributed to the improvement of our research. Your acknowledgement of our work fills me with a sense of accomplishment and honor. I truly appreciate the time and effort you dedicated to reviewing our manuscript.

Once again, thank you for your support and recognition.

With warm regards

Reviewer 3 Report

Comments and Suggestions for Authors

Dear Authors,

thanks a lot for considering many of the suggestions were provided in the former revision. However, still more work should be done for a proper publication of your research outputs. Hence, below you may find further comments and suggestions.

Kind regards

Comments

Line 90-93: the sentence is not clear and with some term repeated (e.g., twelve cages). Moreover, according to this version of the manuscript, each treatment was replicated 4 time; then you can write that.

Line 111: in the previous version of the manuscript, it was stated that the pellet was dried at 90 °C, in the present one that the pellet was dried “open air”. This is quite unusual and even though 90 °C for 20 minutes was just not properly reported, still environmental temperature and environmental humidity, in addition to the drying process duration, should be reported.

Line 115 (Table 1): refer “[Ca (H2PO4 )2]” asMonocalcium phosphate”

Line 127: Is the terms “pregnancy” and “belly” suitable when referring to prawns?!? Please, revise them using more scientifically appropriate terms.

Line 235: in the “note” of table 2, it is still mentioned that the groups were triplicated while in M&M (lines 90-93), it is explained that each groups was replicated 4 time (12 tanks / 3 treatments = 4 tanks).

By the way, to enable the reader to study the data variability, it is necessary to report for each data set either the number of observation, the standard deviation (or the standard error) and the p value of the comparison, or the Standard Error Mean (SEM) and the p value. Please, provide each table of the mentioned information.

Yet, the standard deviation of SR values is still given with the same number of decimal digit of its mean. The other SD values are not always correctly given and they need to be carefully revised.

Line 255 (Table 3): SD of the moisture value of the treatment A, is given with an extra decima digit. Please, revise carefully. Moreover, all the comments given for line 235 must be considered also for table 3 (see Line 235: the number of observations (n), the standard deviation (or the standard error) and the p value of the comparison, or the Standard Error Mean (SEM) and the p value; and number of tanks per each treatment).

Line 299 (Figure 2): when no statistically significant differences are detected, there is no need to show the letter. The absence of different letters automatically implies no differences.

Line 311 (Table 4): see comments given for line 235.

Moreover, the values give for total protein and lipids seems not congruent. In the title of the table it is referred that the data are expressed as mg/g while in the table as mg/kg. Which one is correct. Anyway, “total protein” were 35.81 mg/kg (treatment A) and it seems to be not correct; in fact, in this case protein would be 0.003581 %; also in case the values are given as mg/g, total proteins would be 3,581, still to low protein concentration for egg of prawns. Please, how do you explain that? There is some other mistake? Maybe the data are not given on dry weight but rather on wet weight? Please, check the given data carefully and provide the correct unit.

Comments on the Quality of English Language

Typos are present throughout the manuscript. It requires an appropriate revision. Moreover, sometime not scientifically appropriate terms (e.g., "pregnancy" in the case of prawn) are used and it is higly suggested to improve them.

Author Response

Dear reviewer,

We would like to express our sincere gratitude for the time and effort you dedicated to reviewing the previous version of our manuscript. Your valuable suggestions have significantly contributed to the improvement of our work. We have carefully considered and incorporated your constructive comments and suggestions into the second revised manuscript. We first accepted the initial revision, and then revised it again based on instructions provided in your letter. Accordingly, we have uploaded a copy of the first revised manuscript with all changes highlighted by using the track changes mode in MS Word. In the even that we missed any one of the comments please let me know.

We have provided a point-by-point response to your comments below in red color. Please find the following detailed responses to your comments and suggestions. We sincerely hope that this revised manuscript has addressed all your comments and suggestions. We appreciate your warm work earnestly and hope that the correction will meet with approval.

Lastly, we revised the whole manuscript carefully to avoid language errors. We believe that the language is now acceptable for the review process. Once again, thank you very much for your comments and suggestions. Because of this, our manuscripts will become more scientifically accurate, structurally complete, and logical. We hope that the revised manuscript will meet with approval. If any further action is needed, please let us know immediately. We look forward to hearing back from you.

Sincerely

  • Line 90-93: the sentence is not clear and with some term repeated (e.g., twelve cages). Moreover, according to this version of the manuscript, each treatment was replicated 4 time; then you can write that.

RESPONSE: We sincerely appreciate the valuable suggestions. As suggested, we have rewritten this sentence.

  • Line 111: in the previous version of the manuscript, it was stated that the pellet was dried at 90 °C, in the present one that the pellet was dried “open air”. This is quite unusual and even though 90 °C for 20 minutes was just not properly reported, still environmental temperature and environmental humidity, in addition to the drying process duration, should be reported.

RESPONSE: We feel great thanks for your professional review work on our article. Your comments are really thoughtful. The fact is that we dry the raw diet indoors. Furthermore, considering the effects of light on dietary folic acid contents, we stored the prepared diet in brown ziplock bags in the refrigerator at 4℃ for a long time and only took out an appropriate amount of feed every day. Finally, according to the final measured value, we found that the dietary folic acid levels in each treatment were within a reasonable range.

However, the reason why we wrote “90℃ for 20 min” in the previous manuscript is that our research group has prepared various feeds for a long time, so we did not change the content of this part due to carelessness in referring to the published article on diet preparation. We are very sorry for our carelessness and awkwardness, thank you once again for your valuable contribution to our study.

  • Line 115 (Table 1): refer “[Ca (H2PO4 )2]” asMonocalcium phosphate”
  • Line 127: Is the terms “pregnancy” and “belly” suitable when referring to prawns?!? Please, revise them using more scientifically appropriate terms.

RESPONSE: For the above two suggestions, we sincerely thank you for your careful reading. As suggested, we have corrected the “Ca (H2PO4 )2” into “Monocalcium phosphate”, “pregnancy” into “spawning”, and “belly” into “abdomen”.

  • Line 235: in the “note” of table 2, it is still mentioned that the groups were triplicated while in M&M (lines 90-93), it is explained that each groups was replicated 4 time (12 tanks / 3 treatments = 4 tanks).
  • Line 311 (Table 4): see comments given for line 235.

RESPONSE: For the above two suggestions, we sincerely thank you for your careful reading. Firstly, we feel very sorry for our carelessness. We have revised in our resubmitted manuscript. In fact, the reason why we made this mistake is that in the articles we have published, we often used three replicates, but this time we used four replicates, and we have been using the previous writing method, so we did not correct this mistake, we are very sorry for our mistake.

  • By the way, to enable the reader to study the data variability, it is necessary to report for each data set either the number of observation, the standard deviation (or the standard error) and the p value of the comparison, or the Standard Error Mean (SEM) and the p value. Please, provide each table of the mentioned information.
  • Moreover, all the comments given for line 235 must be considered also for table 3 (see Line 235: the number of observations (n), the standard deviation (or the standard error) and the p value of the comparison, or the Standard Error Mean (SEM) and the p value; and number of tanks per each treatment).

RESPONSE: We sincerely appreciate the valuable suggestions. As per your kind suggestions, we reported the related data including the number of observation, p value, and number of tanks per each treatment.

  • Yet, the standard deviation of SR values is still given with the same number of decimal digit of its mean. The other SD values are not always correctly given and they need to be carefully revised.
  • Line 255 (Table 3): SD of the moisture value of the treatment A, is given with an extra decima digit. Please, revise carefully.

RESPONSE: Thank you very much for your reminder, we have carefully revised these parts, and we have also carefully proofread the whole manuscript.

  • Line 299 (Figure 2): when no statistically significant differences are detected, there is no need to show the letter. The absence of different letters automatically implies no differences.

RESPONSE: We feel very sorry for our carelessness. In our resubmitted manuscript, we have revised this figure.

  • Moreover, the values give for total protein and lipids seems not congruent. In the title of the table it is referred that the data are expressed as mg/g while in the table as mg/kg. Which one is correct. Anyway, “total protein” were 35.81 mg/kg (treatment A) and it seems to be not correct; in fact, in this case protein would be 0.003581 %; also in case the values are given as mg/g, total proteins would be 3,581, still to low protein concentration for egg of prawns. Please, how do you explain that? There is some other mistake? Maybe the data are not given on dry weight but rather on wet weight? Please, check the given data carefully and provide the correct unit.

RESPONSE: We sincerely thank you for your careful reading, but we have carefully checked the use of our units, where mg/g in the title is the unit of egg nutrition, and mg/kg in the table is the unit of folic acid content in the diet. In addition, as to whether the egg nutrition used is dry weight or wet weight you mentioned, we did use dry weight in this experiment, and you think that the protein content is low. However, according to the relevant references we have consulted, the previous studies showed that the protein content of the eggs in this prawn is not much different from the results of the current study.

Reference:

Wang W, Li L, Huang X, et al. Effects of dietary protein levels on the growth, digestive enzyme activity and fecundity in the oriental river prawn, Macrobrachium nipponense. Aquaculture Research, 2022(7):53.DOI:10.1111/are.15803.

Li L, Wang W, Yusuf A, et al. Effects of dietary lipid levels on the growth, fatty acid profile and fecundity in the oriental river prawn, Macrobrachium nipponense. Aquaculture Research, 2020, 51.DOI:10.1111/are.14539.